

# Morpho-biochemical characterization and molecular marker based genetic diversity of pearl millet (*Pennisetum glaucum* (L.) R. Br.)

Darshanaben F. Gunguniya[1], Sushil Kumar[1], Mukesh P. Patel[2], Amar A. Sakure[1], Rumit Patel[1], Dileep Kumar[3] and Vikas Khandelwal[4]

[1] Department of Agricultural Biotechnology, Anand Agricultural University, Anand, Gujarat, India
[2] Agriculture and Horticulture Research Station, Anand Agricultural University, Khambholaj, Gujarat, India
[3] Micronutrient Research Centre, Anand Agricultural University, Anand, Gujarat, India
[4] Plant Breeding, ICAR-All India Coordinated Research Project on Pearl Millet, Mandor, Rajasthan, India

## ABSTRACT

Pearl millet is a key food for millions living in semi-arid and arid regions and is a main diet for poorer populations. The genetic diversity existing in the pearl millet germplasm can be used to improve the micronutrient content and grain yield. Effective and organized exploitation of diversity at morphological and DNA levels is the strategy for any crop improvement program. In this study, the genetic diversity of 48 pearl millet genotypes was evaluated for eight morphological traits and eleven biochemical characters. All genotypes were also characterized using twelve SSR and six SRAP markers to evaluate genetic diversity. The significant mean difference between morphological and biochemical traits were detected. The productive tillers per plant varied from 2.65 to 7.60 with a mean of 4.80. The grain yield of genotypes varied more than $3\times$ from 15.85 g (ICMR 07222) to 56.75 g (Nandi 75) with an average of 29.54 g per plant. Higher levels of protein, iron, and zinc contents were found to be present in ICMR 12555 (20.6%), ICMR 08666 (77.38 ppm), and IC 139900 (55.48 ppm), respectively, during the experiment. Substantial variability was observed for grain calcium as it ranged from 100.00 ppm (ICMR 10222) to 256.00 ppm (ICMR 12888). The top eight nutrient-dense genotypes flowered in 34–74 days and had 5.71–9.39 g 1,000 grain weight. Genotype ICMR 08666 was superior for Fe, Zn, K and P. The inter-genotype similarity coefficient at the genetic level, generated using DNA markers, ranged from 0.616 to 0.877 with a mean of 0.743. A combination of morpho-biochemical traits and DNA markers based diversity may help to differentiate the genotypes and diverse genotypes can be used in breeding programs to improve the mineral content in pearl millet.

Corresponding author
Sushil Kumar,
sushil254386@yahoo.com

## INTRODUCTION

Pearl millet (*Pennisetum glaucum* (L.)) is a small-seeded C4 plant of the Poaceae family. The 1.7 GB genome of this crop is accommodated by 2n=2x=14 chromosomes. Compared to other cereals like wheat and rice, it can effectively withstand drought, nutrient-depleted soil, and hot environmental conditions of the hostile deserts of India and Africa. This hardy nature makes pearl millet resilient to harsher climatic conditions consequently it is cultivated in marginal environments of arid and semi-arid tropical regions of India and south east Asia, sub-Saharan Africa and much of southern and eastern Africa (*Ramya et al., 2018*). Pearl millet is a key food for millions living the semi-arid and arid regions. The grain is mainly consumed as human food while biological yield is used as livestock feed. Pearl millet is a primary food for mankind living in dryland agriculture regions.

Pearl millet accounts for circa 50% of the total global millet production. It is grown on >28 million ha, mainly in Africa and India. India is the world's largest producer of millets, harvesting 11 million tonnes annually, or around 36% of global production. In 2020, India harvested 8.61 million tonnes of pearl millet grains from a 6.93 million ha area with 1,243 kg/ha of productivity (*Directorate of Millets Development, 2020*).

In any breeding strategy, variation continues to be the key to success. Pearl millet shows abundant phenotypic variability for most of the quantitative traits like flowering time, ear head length, grain characteristics, tolerance to various (a)biotic stresses as well as nutritional quality (*Bhattacharjee et al., 2007*). Effective and logical utilization of this diversity is vital to any breeding program (*Allard, 1960*). Exploiting this genetic diversity in the pearl millet population may allow the improvement of micronutrient density in grain and grain yield.

Micronutrient insufficiency has emerged as a global problem, particularly for those living in underdeveloped nations and consuming carbohydrate-rich cereal-based diets. This deficiency can be managed with pearl millet, a nutritious cereal (*Kumar et al., 2016*). Among all coarse cereals, pearl millet grains are dense in minerals like iron (Fe) and zinc (Zn) concentration and essential amino acids. The protein in pearl millet ranges between 9–21%, which is higher than sorghum (10.4%), rice (6.8%), and maize (4.7%) (*Kaur et al., 2014*). The grains of pearl millet are gluten-free and have a low glycemic index due to their high fiber content. The provitamin-A enriched grains are also a richer source of fat (5–7 g/100 g) but are scarce in vitamins B and C (*Gopalan, Rama Sastri & Balasubramanian, 2003*). Pearl millet grain is encased in a tough fibrous seed that contains a variable amount of inhibitory factors like phytic acid and polyphenols (*Arora, Sehgal & Kawatra, 2003*). However, these factors can be reduced through various approaches like soaking, fermentation, blanching, and roasting (*Kaushik & Grewal, 2017*) up to a certain extent only. Moreover, the presence of anti-nutrient factors like saponins, tannins and phytic acid which can reduce nutrient utilization or food uptake hinders the biofortification in millet.

To improve the nutritional quality and diminish the anti-nutritional factors of pearl millet through any breeding approach, knowledge about the variability for mineral

content, anti-nutritional factors and their relation with yield is a prerequisite. Accumulation of both micro- and anti-nutrients in seeds is a complex mechanism containing numerous genes and affected by the environment (*Anuradha et al., 2017a*; *Anuradha et al., 2017b*). Due to the confounding effect of the environment, similar genotypes would have different phenotypes due to environmental variability. Therefore, it is hard to equate morpho-biochemical and genetic variability. In contrast, molecular markers reflect the authentic genetic variability and relationships among accessions than phenotypic markers (*Glaszmann et al., 2010*). In pearl millet, microsatellite, single nucleotide polymorphisms (SNP) and restriction fragment length polymorphism (RFLP) markers have been applied to create linkage maps followed by quantitative trait loci mapping and germplasm characterization (*Kumar et al., 2020a*; *Kumar et al., 2020b*). The density and genome coverage can be improved by the combination of various markers. In pearl millet, no report is available on the deployment of sequence-related amplified polymorphism (SRAP) markers for genetic diversity assessment though SRAP has been used for linkage map development by *Pedraza-Garcia, Specht & Dweikat (2010)*. Therefore, in the current experiment, both SRAP and microsatellite markers have been used to expose the genetic diversity. SRAP markers are dominant markers that target genomic coding sequences and have been employed for genetic diversity assessment (*Li & Quiros, 2001*). The current study was created to analyse the natural variability for grain mineral and anti-nutrient content in grain as well as molecular diversity in pearl millet germplasm with the objective of improving the nutritional quality and food safety of pearl millet as well as expanding understanding in this area.

# MATERIALS AND METHODS

The field trial was done in a randomized complete block design (RCBD) with two replications. The inter- and intra-row distance was 60 and 15 cm, respectively. The recommendations for crop management practices were followed for uniform plant growth and a healthy crop stand. The seeds were sown on February 2021. A total of 48 pearl millet genotypes were used for the study (Table S1).

## Morphological characters

The panicles were covered with glassine bags to prevent cross-pollination by outside pollen and to collect self-seeds. For phenotyping of grain based traits, physiologically mature panicles were collected, dried under sunlight, and then manually threshed in bulk.

The experimental material was evaluated for eight morphological traits viz., days to 50% flowering, plant height, panicle diameter (PD, cm), panicle length (PL, cm), number of productive tillers, grain yield, days to maturity, and 1,000 grain weight. Except for days to 50% flowering and days to maturity, which were recorded on a plot basis during the study, data on the above traits were collected from randomly tagged five competitive plants in each genotype in both replications. PD was measured with Vernier calipers.

## Biochemical characters

Before biochemical analysis, grains were cleaned followed by hot air oven drying (80 °C for 24 h). Dried grains (10 g) were powdered manually. For mineral analysis, 0.5 g flour

was processed in 11 mL of nitric acid (69%) and one mL of $H_2O_2$. The digestions were carried out in HVT50 vessels using rotor 12HVT50 in Multiwave GO/Multiwave GO plus (Anton Paar GmbH, Graz, Austria). For microwave digestion the initial temperature was kept at 180 °C with a ramp of 20 min and a hold period of 12 min. While the second round of digestion was performed with a ramp time of 10 mins at 70 °C temperature and a holding time of 5 mins. Inductively Coupled Plasma Optical Emission Spectrometry (ICP-OES (model 7000DV; Perkin Elmer, Waltham, MA, USA, wintab32 software ver. 5.1)) was used to determine the mineral content (iron (Fe), zinc (Zn), calcium (Ca), copper (Cu), and manganese (Mn)) in grain after diluting the digested mixture to a volume of 50 ml using distilled water. The flow rate in a peristaltic pump was 1.5ml per min. From the acid extract, potassium (K) content was quantified using a flame photometer (*Jackson, 1973*), while the vanadate-molybdate method of *Jackson (1973)* was used to estimate the phosphorus (P). Total phenols were estimated using the Folin-Ciocalteau reagent as *Malik & Singh (1980)* and reading was measured at 730 nm using a spectrophotometer. Soxhlet extraction was performed to estimate the crude oil content, and semimicro-Kjeldahl was employed to determine the crude protein content.

## Molecular marker study

Genomic DNA was extracted from tender leaves as per *Mace et al. (2003)*. Genotyping was done using simple sequence repeat (SSR) and sequence-related amplified polymorphism (SRAP) markers. For SSR marker profiling, markers from the PMES series (*Zala et al., 2017*) were amplified in SensoQuest Thermocycler (Göttingen, Germany). The SSR-PCR reaction conditions were as follows: 94 °C (initial denaturation) for 5 min., followed by 35 cycles of 94 °C for 45 s, X°C (primer specific) for 45 s, 72 °C for 45 s, and 72 °C for 7 mins (final extension). The SRAP amplification was as follows: 94 °C (initial denaturation) for 5 mins, followed by 5 cycles of 94 °C for 30 s, 35 °C for 45 s, and 72 °C for 90 s, followed by 35 cycles of 94 °C for 30 s, X°C (primer specific) for 45 s, 72 °C for 60 s and 72 °C for 10 min (final extension). An agarose gel (3%) was used to resolve PCR products.

## Statistical analysis

The mean value of traits was figured out, and analysis of variance (ANOVA) was performed in accordance with *Panse & Sukhatme (1978)* in Microsoft Excel 2013. A phenotypic trait-based dissimilarity matrix was constructed using Manhattan coefficients with Numerical Taxonomy and Multivariate Analysis System (NTSYSpc 2.0; *Rohlf, 1998*). The amplified products of SSR and SRAP markers were scored in 1 (presence) and 0(absence) fashion. Polymorphism information content (PIC), Multiplex ratio (MR), effective multiplex ratio (EMR) marker index (MI) and resolving power (Rp) value were estimated following *Sharma et al. (2016)* in Microsoft Excel 2013. In NTSYSpc 2.0, The SIMQUAL program used Jaccard's similarity (J) coefficient to compute the genetic similarity between genotypes, SAHN clustering method was used to construct the unweighted pair group method with arithmetic mean (UPGMA) dendrogram (*Sneath & Sokal, 1973*).

**Table 1  Analysis of variance (ANOVA) of studied traits in pearl millet.**

| Trait | Source of variation and mean squares | | |
|---|---|---|---|
| | Replication ($df = 1$) | Genotypes ($df = 47$) | Error ($df = 47$) |
| Days to 50% flowering | 0.167 | 322.311[*] | 5.954 |
| Plant height | 66.833 | 917.408[*] | 62.211 |
| Head diameter | 0.027 | 0.159[*] | 0.027 |
| Panicle length | 1.927 | 46.955[*] | 3.880 |
| Productive tillers per plant | 0.027 | 3.712[*] | 0.217 |
| Grain yield per plant | 11.003 | 203.764[*] | 30.441 |
| Days to maturity | 2.344 | 9.292[*] | 3.471 |
| 1000 grain weight | 0.000176 | 2.863[*] | 0.104 |
| Protein content | 2.004 | 10.363[*] | 0.705 |
| Lipid content | 0.062 | 3.851[*] | 0.052 |
| Iron content | 14.015 | 276.360[*] | 16.714 |
| Zinc content | 41.12 | 79.610[*] | 13.100 |
| Manganese content | 2.154 | 11.232[*] | 1.576 |
| Calcium content | 661.5 | 2061.205[*] | 332.691 |
| Copper content | 0.218 | 43.735[*] | 0.551 |
| Potassium content | 57.722 | 102.837[*] | 14.517 |
| Phosphorus content | 5.782 | 12.004[*] | 5.093 |
| Phytate content | 895.482 | 8396.470[*] | 541.246 |
| Total phenolic acid | 26.471 | 112.792[*] | 36.582 |

**Notes.**
*Significant at 5% level of probability.

# RESULTS AND DISCUSSION

The ANOVA resulted that genotypic variations were significant at a 5% level of probability for all the traits, showing ample genetic diversity among the genotypes under study (Table 1). This also suggested that there is sufficient scope to select superior breeding material which can be exploited in pearl millet breeding programs.

## Character variance analysis
### Morphological parameters

Early flowering is a desirable trait for pearl millet as it is a crop of semi-arid and arid regions. Earliness becomes an important trait in areas where scanty and erratic rains aggravate the moisture stress condition during the growth stage of the crop and leads to post-flowering moisture stress (*Yadav et al., 2011*). In the current study, though, the population mean for days to 50% flowering was 54.69 days but the days to 50% flowering ranged from 34 (IC 370523) to 77 days (ICMR 07999). Earlier literature also recorded similar values for days to 50% flowering for example 49.06 days by *Govindaraj et al. (2011)*, 53.10 days by *Sonali et al. (2019)* and 55.61 days by *Pallavi et al. (2020)*. PH is an important trait that governs tradeoffs between competition and resource distribution, which is decisive for productivity (*He et al., 2021*). Semi-dwarf genotypes are better than their tall counterparts because of their reduced lodging vulnerability and better response

to nitrogen (*Azhaguvel et al., 2003*). In the present experiment, PH ranged from 110.10 cm (ICMR 06555) to 205.35 cm (IC 332715) with an average of 149.42 cm. The results indicated that most of the studied genotypes are semi-dwarf in nature and with better management dwarfism supports the grain yield.

Panicle size (length and diameter) are two important traits that have direct positive correlations with grain yield in pearl millet (*Vengadessan et al., 2013*). Hence, the improvement of sink-size linked traits is a key objective for pearl millet improvement programs. PL in the present study ranged from 15.55 cm (ICMR 11888) to 38.05 cm (IC-332716) with an average of 24.03 while the diameter ranged from 1.03 cm (ICMR 10999) to 2.15 cm (ICMR 09333) with an average of 1.53 cm. *Abubakar et al. (2019)* observed a similar range and mean in pearl millet (2.26 cm) for panicle diameter. Similarly, results for PL are comparable with previous reports (*Sharma et al., 2018*; *Rani et al., 2019*). The number of productive tillers per plant varied from 2.65 (ICMR 08222, ICMR 11999) to 7.60 (IC 370523) with an average of 4.80 (Table 2). According to *Siles et al. (2004)*, non-tillering millet genotypes produced bold seeds having TGW >10 gm than the genotype that produced tillers. Similarly, *Maman et al. (2004a)* and *Maman et al. (2004b)* also reported that, a reduction in productive tillers from 10 to three or five improved seed yields by 15–30%. *Yadav et al. (2021)* reported that private-sector hybrids are generally have less effective tillers/plant. However, farmers in drought-prone areas prefer high tillering hybrids because tillering is a strategy of adaptation to intermittent drought spells (*Yadav et al., 2016*).

In cereal breeding, yield, a complex trait, is one of the supreme traits which is influenced by several associated traits. The grain yield of genotypes varied more than $3\times$ from 15.85 g (ICMR 07222) to 56.75 g (Nandi 75) with an average of 29.54 g per plant. Large variability was also observed for 1000 grain weight (TGW) which is determined by the form, size and density of the grain and these are directly related to total grain yield. TGW ranged from 4.93 g (ICMR 06888) to 10.45 g (ICMR 06555) with an average of 7.14 g. A diversity assessment of 21,594 pearl millet genotypes from 50 nations revealed huge variability for the TGW (1.5 to 21.3 g) (*Upadhyaya, Reddy & Gowda, 2007*). Three-fold variability for TGW (6–16 g) was earlier recorded by *Pujar et al. (2018)*.

### Biochemical parameters

Compared to other main cereal crops, pearl millet yields more nutritious grains with protein, Ca, P, Fe, and Zn (*Devos, Hanna & Ozias-Akins, 2006*). Currently, the commercially grown varieties/hybrids of pearl millet produce grains with an average Fe and Zn content of 42 and 32 ppm (parts per million), respectively (*Rai et al., 2016*). However, a much wider variability for these micro-nutrients has been reported in germplasm collections (*Rai et al., 2014*). Fe is an essential element for blood production and for the growth and development of the body. Zn is essential for the development of a strong immune system. The values of Fe content in the current study ranged from 31.58 (ICMR 07777) to 77.38 (ICMR 08666) with an average of 49.69. Zn content ranged from 29.34 (ICMR 08111) to 55.48 (IC 139903) with an average of 39.36 (Table 2). A similar mean value was observed by *Velu et al. (2007)* where grain Fe was 45.50 ppm. In previous studies, values

**Table 2  Descriptive statistics for studied traits in pearl millet.**

| Trait | Mean | Range | S.Em | CD @ 5% | CV% |
|---|---|---|---|---|---|
| Days to 50% flowering | 54.69 | 34.00 (IC-370523)–77.00 (ICMR 07999) | 1.73 | 4.91 | 4.46 |
| Plant height (cm) | 149.42 | 110.10 (ICMR 06555)–205.35 (IC-332715) | 5.58 | 15.87 | 5.28 |
| Head diameter (cm) | 1.53 | 1.03 (ICMR 10999)–2.15 (ICMR 09333) | 0.12 | 0.33 | 10.75 |
| Panicle length (cm) | 24.03 | 15.55 (ICMR 11888)–38.05 (IC-332716) | 1.39 | 3.96 | 8.2 |
| Productive tillers per plant | 4.80 | 2.65 (ICMR 08222, ICMR 11999)–7.60 (IC-370523) | 0.33 | 0.94 | 9.7 |
| Grain yield per plant | 29.54 | 15.85 (ICMR 07222)–56.75 (Nandi 75) | 3.9 | 11.1 | 18.67 |
| Days to maturity | 85.64 | 80.50 (ICMR 12333)–89.00 (ICMR 08999, ICMR 07777) | 5.82 | 16.57 | 9.71 |
| 1000 grain weight | 7.14 | 4.93 (ICMR 06888)–10.45 (ICMR 06555) | 0.23 | 0.65 | 4.51 |
| Protein content (%) | 13.73 | 8.26 (AICRP-PM-12)–20.06 (ICMR 12555) | 0.59 | 1.68 | 6.12 |
| Lipid content (%) | 4.68 | 2.72 (ICMR 06999)–6.95 (ICMR 08444) | 0.16 | 0.46 | 4.89 |
| Iron content (ppm) | 49.69 | 31.58 (ICMR 07777)–77.38 (ICMR 08666) | 2.89 | 8.22 | 8.23 |
| Zinc content (ppm) | 39.36 | 29.34 (ICMR 08111)–55.48 (IC 139903) | 2.65 | 7.53 | 9.5 |
| Manganese content (ppm) | 14.04 | 7.20 (ICMR 07222)–17.63 (ICMR 08444) | 0.89 | 2.53 | 8.94 |
| Calcium content (ppm) | 199.31 | 100.00 (ICMR 10222)–256.00 (ICMR 12888) | 12.9 | 36.69 | 9.15 |
| Copper content (ppm) | 9.86 | 4.92 (AICRP-PM- 6)–22.59 (GHB 558) | 0.52 | 1.49 | 7.52 |
| Potassium content (ppm) | 4798 | 1,800 (ICMR 07222)–6,020 (ICMR 10999) | 2.69 | 7.67 | 7.94 |
| Phosphorus content (ppm) | 3112 | 2,258 (AICRP-PM-62)–3,672 (ICMR 08666) | 1.6 | 4.54 | 7.25 |
| Phytate content (mg/100 g) | 282.39 | 201.5 (ICMR 08111)–542.5 (GHB 558) | 16.45 | 46.8 | 8.24 |
| Total phenolic acid (mg/100 g) | 60.26 | 75.16 (ICMR 12555)–44.41 (Nandi 75) | 4.28 | 12.17 | 10.04 |

**Notes.**

S.Em., Standard error of mean; CD @ 5%, critical difference at 5% level of significance; CV, Coefficient of variance.

ranging from 45.50–55.73 for grain Fe concentration and from 38.60–46.61 for grain Zn concentration have been reported (*Anuradha et al., 2017a*; *Anuradha et al., 2017b*; *Anuradha et al., 2018a*; *Anuradha et al., 2018b*; *Sonali et al., 2019*).

In the human body, fats and carbohydrates metabolism, absorption of Ca, and the control of blood sugar are all impacted by manganese. It is also essential for standard brain/nerve functioning and bone mineral density. The values of Mn content ranged from 7.20 ppm (ICMR 07222) to 17.63 ppm (ICMR 08444) with an average of 14.04 ppm. The outcome is in congruence with *Anuradha et al. (2017a)*, *Anuradha et al. (2017b)*, *Kumar et al. (2020a)* and *Govindaraj et al. (2020)*. Similarly, a low value of Mn (8 ppm) was recorded by *Oshodi, Oqungbenle & Oladimeji (1999)*.

Ca is very important for the contraction of muscle; the development of strong bones and teeth, blood clotting, the transmission of nerve impulses, and in the regulation heart beats. This is claimed that a high intake of cereal grains increases the chances of Ca deficiency. However, this is not true with pearl millet as substantial variability was observed for grain Ca as it ranged from 100.00 ppm (ICMR 10222) to 256.00 ppm (ICMR 12888) with an average of 199.31 ppm. Higher variability for Ca (85–249 ppm) was also recorded by *Govindaraj et al. (2020)* in pearl millet core collection. In the current study, 50% of the genotypes had high Ca (>200 ppm).

Cu is essential for the synthesis of elastin and collagen. It is a key cofactor of many metalloenzymes playing role in metabolism Fe and cellular respiration. In the current

study, the grain Cu ranged from 4.92 ppm (IC 332703) to 22.59 ppm (GHB 558) with an average of 9.86 ppm. The range of grain Cu in various studies is different like 4.14–15.35 ppm in *Anuradha et al. (2018a)*; *Anuradha et al. (2018b)*, 4–7 ppm in *Govindaraj et al. (2020)*, 3.19–4.76 ppm in *Warrier et al. (2020)*.

The transport of water, nutrients, and carbohydrates within plant cells is linked with potassium. It is a crucial mineral for the activation of several enzymes that control the synthesis of protein, starch, and adenosine triphosphate (ATP) in plants. The K ranged from 1,800 ppm (ICMR 07222) to 6,000 ppm (ICMR 10999) with an average of 4,700 ppm. Large variability for K was also recorded in 122 commercial pearl millet cultivars (3,675–5,375 ppm; *Govindaraj et al., 2020*) and core collection (3,667–5,133 ppm; *Govindaraj et al., 2020*).

The body needs P to produce protein for the development, upkeep, and repair of cells and tissues. Additionally, it participates in the production of ATP. The values of P content ranged from 2,200 (IC 139900) to 3,600 ppm (ICMR 08666) with an average of 3,112 ppm. ICMR 06555 was statistically at par with IC 139900.

Pearl millet is also a promising source of protein. Studies indicated that protein in pearl millet is circa 11.8%, which is better than rice (8.6%), and maize (9.2%) and comparable with sorghum (10.7%). Moreover, pearl millet grain is enriched with glutamate which is a precursor of $\gamma$-aminobutyric acid (GABA) (*Tomar et al., 2021*). In the current study the protein content ranged from 8.26% (IC 332716) to 20.06% (ICMR 12555) with an average of 13.73%. ICMR 07444 (9.89%) was statistically at par with IC 332716. The study of *Pujar et al. (2020)* reported grain protein content variation between 6–18%, with a mean of 11%. The augmentation of pearl millet in daily food can reduce the risk of protein malnutrition in an economical way. Moreover, protein extracted from pearl millet can be exploited to design protein-enriched functional foods.

The lipid content ranged from 2.72% (ICMR 06999) to 6.95% (ICMR 08444) with an average of 4.68%. A comparable range and mean were observed by *Arulselvi et al. (2007)* (5.12%), *Abdalla et al. (1998)* (2.70–7.10%) and *Tomar et al. (2021)* (5.24–9.99). The lipid content in pearl millet ranges from 1.5 to 6.8% which is higher sorghum and other millets (*Hassan, Sebola & Mabelebele, 2021*). Though, the high lipids have been documented as possible causes for the rancidity of millet flour. However, the shelf life of flour can be increased by hydrothermal treatment, irradiation, cooling storage, or a combination of more than one technology (*Goyal & Chugh, 2017*).

The metal-chelating ability of phytic acid makes it is an antinutritional phytochemical as it declines the bio-availability of ions like Mn, Ca, Mg, Fe and Zn (*Marathe et al., 2018*). In the current study, the phytate ranged from 201.5 mg/100 g (ICMR 08111) to 542.50 mg/100 g (GHB 558) with an average of 282.39. *Abdalla et al. (1998)* also recorded a similar range from 354–795 mg/100 g of phytate. *Gabaza et al. (2018)* reported that phytate in pearl millet grains ranges between 580 mg/100 g to 1,380 mg/100 g which is similar to sorghum and maize. The range of phytate in the current study is supported by the result of *Pushparaj & Urooj (2014)* in Indian cultivars where it was between 0.26–0.99 g/100 g. The result suggested that phytic acid content in pearl millet grain is significantly

lower than in rice (0.68–1.03%; *Liu, 2005*), oat (0.5–1.2%; *Peterson, 2001*), soybean (1.0–2.22%; *Lolas, Palamidids & Markakis, 1976*) and wheat (0.2–2.9%; *Gupta, Gangoliya & Singh, 2015*) Hence, regular consumption will possibly not hamper the bioavailability of minerals.

Polyphenols have many health benefits as having antioxidant activity. Moreover, phytic acid is considered to be beneficial in dropping cholesterol and reducing cancer risk. The values of total phenolic acid ranged from 44.41 mg/100 g (Nandi 75) to 75.16 mg/100 g (ICMR 12555) with an average of 60.26 mg/100 g. Higher phenol content in grain makes pearl millet a good food to maintain the redox potential of cells and to quench the ROS species. Phenolic may be particularly important in the treatment of postprandial hyperglycemia since it has been documented that it reduces intestinal-glucosidase and pancreatic-amylase (*Shobana, Sreerama & Malleshi, 2009*).

## Nutrient-dense genotypes

Genotypes dense in multiple nutrients can directly be released as a variety after evaluating their yield performance over the locations for multiple years. Such genotypes can be exploited in a hybridization program. In the current study, the top eight nutrient-dense genotypes flowered in 34–74 days and had 5.71–9.39 g TGW (Table 3). Top genotypes had Fe content of 61.07–77.38 (ICMR 08666) ppm, Zn content of 45.11–55.48 (IC 139900) ppm, Mn content of 16.2–17.63 (ICMR 08444) ppm, Ca content of 230.5-256 (ICMR 12888) ppm, Cu content of14.32–22.59 (GHB 558) ppm, K content of 52.3–60.15(ICMR 10999) ppm, and P content of 33.28–36.72 (ICMR 08666). IC 139900 was superior for both Fe (71.22 ppm) and Zn (55.48 ppm). Genotype ICMR 08666 was dense for Fe, Zn, K and P. Out of eight high-Fe genotypes, only two genotypes had >75 ppm. Thus current experiment also identified the best genotypes that had a higher content of multiple nutrients. Earlier, *Govindaraj et al. (2020)* also identified genotypes having a high content of multiple nutrients.

## Phenotypic diversity analysis

Phenotypic diversity is important for pearl millet breeding. The interactions between the genome and all of its growing micro- and mega-environments lead to the phenotype of the plant (*Fasoula, Ioannides & Omirou, 2020*). The mean value of each trait was used to generate the Manhattan dissimilarity coefficient and dendrogram (*Sokal, 1958*). The genotypes were divided into nine major clusters based on the Manhattan dissimilarity coefficient. Earlier, *Shashibhushan, Kumar & Kondi (2022)* also generated eight clusters of 40 pearl millet genotypes using phenotypic data (Fig. 1). In current study, the average dissimilarity value among genotypes was calculated to be 0.16, demonstrating modest phenotypic variability (Table 4). The dissimilarity between genotypes ranged from 0.08 (IC 139899 and ICMR 07888) to 0.27 (Nandi 75 and ICMR 07222) for the respective pair of genotypes.

Cluster I comprise seven genotypes, characterized by high values of DFF, days to maturity, lipid, potassium and low values of NPT. Cluster II consists of 25 genotypes. Cluster III contains four genotypes, namely ICMR 10222, Nandi 75, ICMR 12111, and

**Table 3  Top eight genotypes for different nutritional traits with their agronomic performance.**

| Trait | Top 8 genotypes | Range | |
|---|---|---|---|
| | | DFF | TGW (g) |
| Fe | ICMR 08666, ICMR 11888, IC 332727, IC 139900, ICMR 12555, ICMR 06111, ICMR 08333, ICMR 07888 | 37.00–72.00 | 5.91-8.45 |
| Zn | IC 139903, ICMR 08999, ICMR 06222, ICMR 12999, ICMR 10999, ICMR 08666, IC 332715, IC 332703 | 42.00–70.50 | 5.94–9.39 |
| Mn | ICMR 08444, ICMR 12555, ICMR 09888, ICMR 08333, ICMR 06666, ICMR 12666, ICMR 09222, IC 332703 | 45.00–74.50 | 5.91–8.01 |
| Ca | ICMR 12888, ICMR 12777, ICMR 07444, ICMR 09333, ICMR 11777, IC 332703, ICMR 08999, ICMR 08333 | 35.50–72.50 | 5.91–8.36 |
| Cu | GHB 558, IC 332715, IC 332716, ICMR 06666, ICMR 07222, ICMR 07444, ICMR 07777, ICMR 10999 | 42.00–73.50 | 5.71–8.02 |
| K | GHB 558, GHB 732, GHB 905, ICMR 08666, ICMR 10999, ICMR 12555, ICMR 12666, ICMR 12777 | 43.50–74.00 | 6.16–8.99 |
| P | ICMR 08666, ICMR 06888, ICMR 06111, IC 370523, IC 332727, IC 139899, Nandi 75, ICMR 06666 | 34.00–73.50 | 4.93–9.34 |

**Notes.**
DFF, days to 50% flowering; TGW, 1,000-grain weight.

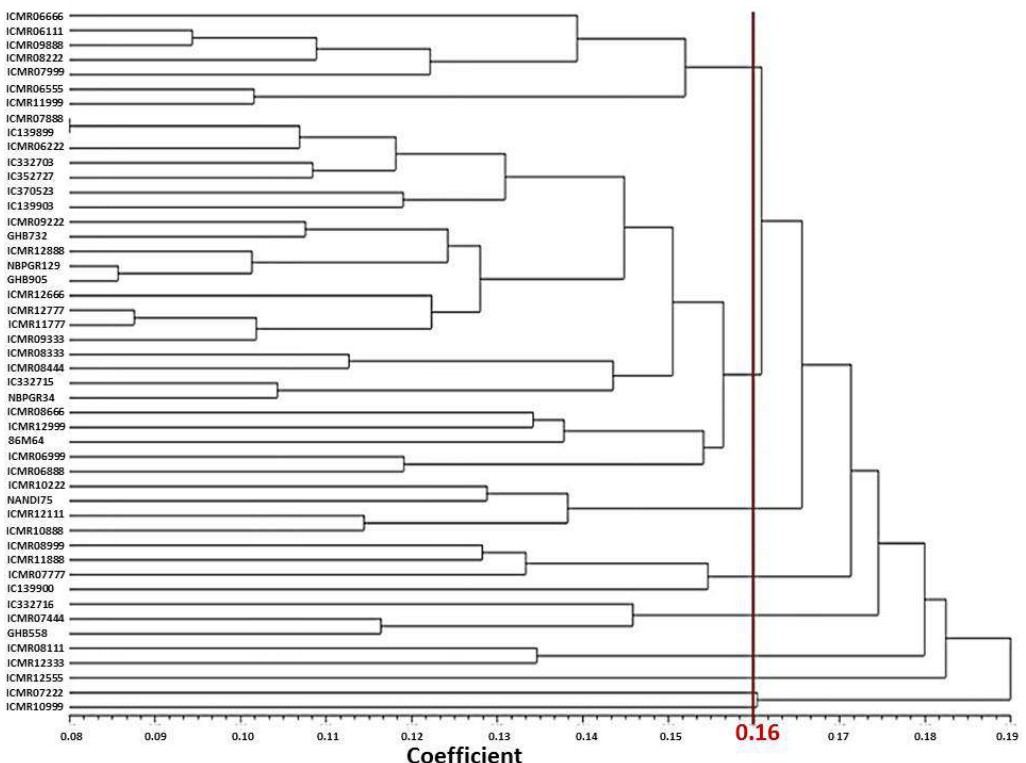

**Figure 1  UPGMA based dendrogram with Manhattan dissimilarity coefficient of studied traits.**

ICMR 10888. This cluster is characterized by more GY, days to maturity and TGW. Cluster IV has four genotypes, namely ICMR 08999, ICMR 11888, ICMR 139900 and

**Table 4 Variability for mean values of 19 quantitative traits in nine groups identified by Manhattan dissimilarity coefficient.**

| Trait | Number of genotypes in each cluster | | | | | | | | |
|---|---|---|---|---|---|---|---|---|---|
| | 7 | 25 | 4 | 4 | 3 | 2 | 1 | 1 | 1 |
| Days to 50% flowering | 72.86 | 50.24 | 50.50 | 44.88 | 57.67 | 62.00 | 53.50 | 56.50 | 70.50 |
| Plant height (cm) | 126.56 | 152.87 | 174.28 | 139.36 | 172.68 | 139.78 | 125.45 | 135.95 | 151.25 |
| Head diameter (cm) | 1.66 | 1.55 | 1.49 | 1.59 | 1.41 | 1.16 | 1.65 | 1.45 | 1.03 |
| Panicle length (cm) | 23.71 | 23.68 | 25.99 | 23.90 | 30.53 | 19.23 | 22.00 | 26.00 | 17.95 |
| Productive tillers per plant | 3.67 | 4.98 | 4.36 | 5.29 | 4.70 | 6.70 | 5.90 | 4.70 | 3.50 |
| Grain yield per plant | 24.72 | 29.63 | 48.99 | 22.53 | 27.23 | 22.33 | 47.75 | 15.85 | 28.35 |
| Days to maturity | 86.50 | 85.12 | 86.13 | 88.38 | 87.00 | 81.50 | 85.00 | 86.00 | 84.00 |
| 1000 grain weight | 7.83 | 7.12 | 7.79 | 6.50 | 7.04 | 6.32 | 7.27 | 5.71 | 6.16 |
| Protein content (%) | 12.50 | 14.49 | 12.72 | 12.61 | 9.71 | 13.69 | 20.06 | 15.72 | 15.71 |
| Lipid content (%) | 5.67 | 4.36 | 4.72 | 5.64 | 3.25 | 4.13 | 2.82 | 6.92 | 6.64 |
| Iron content (ppm) | 52.03 | 50.39 | 37.14 | 55.66 | 46.22 | 43.33 | 69.65 | 48.98 | 46.16 |
| Zinc content (ppm) | 36.59 | 40.25 | 35.21 | 47.05 | 36.07 | 30.54 | 35.13 | 44.98 | 48.79 |
| Manganese content (ppm) | 13.63 | 14.94 | 13.65 | 13.30 | 14.40 | 9.66 | 17.28 | 7.20 | 10.39 |
| Calcium content (ppm) | 191.57 | 208.04 | 144.75 | 207.25 | 215.83 | 158.75 | 229.00 | 204.50 | 218.50 |
| Copper content (ppm) | 9.40 | 8.02 | 7.19 | 13.85 | 19.78 | 9.44 | 8.18 | 17.29 | 19.29 |
| Potassium content (ppm) | 45.70 | 49.33 | 46.08 | 49.31 | 49.00 | 45.58 | 52.30 | 18.00 | 60.15 |
| Phosphorus content (ppm) | 31.43 | 31.81 | 30.68 | 27.54 | 31.98 | 29.23 | 27.75 | 31.52 | 32.07 |
| Phytate content (mg/100 g) | 283.74 | 272.46 | 327.68 | 248.10 | 398.32 | 206.55 | 287.75 | 307.10 | 250.95 |
| Total phenolic acid (mg/100 g) | 58.59 | 58.50 | 58.21 | 66.53 | 65.50 | 59.57 | 75.16 | 56.96 | 73.15 |

ICMR 07777. This cluster is characterized by more lipid content, Mn content, and days to maturity. Cluster V has three genotypes (IC 332716, GHB 558, ICMR 07444) which are characterized by low values of head diameter and lipid content. Cluster VI consists of two genotypes (ICMR 08111 and ICMR 12333). This cluster is characterized by more productive tillers per plant with low content of Zn. Cluster VII has only one genotype (ICMR 12555) which has high values for characters like days to maturity, protein content, Mn content and TGW (Table 4).

## Molecular marker based diversity

Forty-eight genotypes of pearl millet were analyzed using SSRs and SRAP markers (Table 5). During the experiment, a total of 50 SSR markers were screened for amplification. Out of 50 markers, 30 (60%) primers showed amplification. Of these 30 SSR markers, 12 (37.5%) markers were found polymorphic. These 12 polymorphic SSRs markers generated 65 amplicons. The molecular weight of the amplicon ranged from 85 bp (PMES 190) to 292 bp (PMES 171). In previous reports with PMES-series SSR markers, *Zala et al. (2017)* recorded amplicon size from 101 to 285 bp. The number of polymorphic bands/amplicons per SSR marker ranged from 2 to 13, with a mean of 6.71. All SSR amplicons were found polymorphic. The PIC value, the informativeness of a primer, for each marker was computed for the estimation of marker allelic variation considering the allele frequencies in studied genotypes. The mean PIC of SSR markers was 0.28 though

**Table 5  Amplification details of DNA markers.**

| Maker name | Band size (bp) | TB | Polymorphism (%) | PI | PIC | RP | Mean RP |
|---|---|---|---|---|---|---|---|
| **SSR marker system** | | | | | | | |
| PMES153 | 145-154 | 4 | 100 | 1.14 | 0.29 | 2.00 | 0.50 |
| PMES157 | 135-151 | 4 | 100 | 1.46 | 0.37 | 1.96 | 0.49 |
| PMES160 | 134-150 | 4 | 100 | 1.36 | 0.34 | 2.13 | 0.53 |
| PMES161 | 155-233 | 3 | 100 | 1.23 | 0.41 | 2.04 | 0.68 |
| PMES162 | 132-162 | 7 | 100 | 1.35 | 0.19 | 1.63 | 0.23 |
| PMES168 | 222-284 | 10 | 100 | 1.96 | 0.19 | 2.38 | 0.24 |
| PMES170 | 154-195 | 9 | 100 | 1.43 | 0.16 | 2.00 | 0.22 |
| PMES171 | 151-292 | 13 | 100 | 1.71 | 0.13 | 2.00 | 0.15 |
| PMES173 | 216-203 | 8 | 100 | 0.99 | 0.50 | 2.00 | 1.00 |
| PMES185 | 176-200 | 8 | 100 | 1.55 | 0.19 | 1.83 | 0.23 |
| PMES190 | 85-104 | 6 | 100 | 1.53 | 0.26 | 1.92 | 0.32 |
| PMES199 | 213-305 | 4 | 100 | 1.27 | 0.32 | 1.83 | 0.46 |
| Average | | 6.17 | | 1.42 | 0.28 | 1.98 | 0.42 |
| **SRAP marker system** | | | | | | | |
| Em6+Me2 | 205-1234 | 25 | 100 | 8.11 | 0.32 | 13.25 | 0.53 |
| Em2+Me2 | 94-1152 | 34 | 100 | 7.96 | 0.26 | 10.20 | 0.30 |
| Em1+Me2 | 223-1125 | 18 | 100 | 5.33 | 0.30 | 6.88 | 0.38 |
| Em5+Me4 | 120-1020 | 17 | 100 | 3.81 | 0.22 | 4.63 | 0.27 |
| Em6+Me3 | 450-1357 | 10 | 100 | 2.33 | 0.23 | 5.42 | 0.54 |
| Em2+Me3 | 313-1065 | 26 | 100 | 6.57 | 0.25 | 9.25 | 0.36 |
| Average | | 21.67 | | 5.69 | 0.26 | 8.27 | 0.4 |

**Notes.**

TB, Total Number of Bands; PI, Primer index; PIC, Polymorphic Information Content; RP, Resolving Power.

it varied from 0.132 (PMES 171) to 0.499 (PMES 173). This range was comparable with *Zala et al. (2017)* where PIC was between 0.188–0.375.

The Rp was estimated considering the proportion of genotypes containing the amplicon. The primer that might best differentiate the cultivar can easily be identified by the value of the Rp and PIC. In the current study, Rp varied from 1.625 (PMES 162) to 2.375 (PMES 168), with an average of 1.98. Mean Rp was between 0.154 (PMES 171) to 1.000 (PMES 173). The PI ranged from 0.998 (PMES 173) to 1.963 (PMES 168), though the mean PI value was 1.420. MI is considered to be an inclusive measure of the efficiency to detect polymorphism. The SSR MI was 19.092.

In the case of the SRAP markers, of 25 SRAP, six (24%) were polymorphic. The polymorphic SRAPs amplified 119 amplicons. The product size for SRAPs ranged from 94 (Em2+Me2) to 1357 bp (Em6+Me3). The polymorphic bands ranged from 10 (Em6+Me3) to 34 (Em2+Me2), with a mean of 21.67. *Liu et al. (2008)* observed a polymorphic band detected with each ranging from 6 to 17, with an average of 11.76. *Bhatt et al. (2017)* had a band size from 120 to 500 bp in cumin.This suggested that in different crops SRAP amplicon size will be highly variable. PIC oscillated from 0.224

**Table 6  Comparison between SSR and SRAP marker system.**

| Marker system | Total markers | TB | PB | FP | $H_{av}$ | MR | EMR | MI |
|---|---|---|---|---|---|---|---|---|
| SSR | 12 | 74.00 | 74.00 | 1.00 | 3.35 | 5.69 | 5.69 | 19.092 |
| SRAP | 6 | 130.00 | 130.00 | 1.00 | 1.57 | 10.00 | 10.00 | 15.655 |

Notes.

TB, Total Number of Bands; PB, Number Polymorphic Bands; PI, Primer index; PIC, PolymorphicInformation Content; RP, Resolving Power; FP, Fractionation of Polymorphism; Hav, Average PIC; MR, Multiplex Ratio; EMR, Effective Multiplex Ratio; MI, Marker Index.

(Em5+Me4)—0.324 (Em6+Me2), with an average of 0.26. *Bhatt et al. (2017)* reported a PIC value (0.34).

The mean PI of SRAP markers was 5.69, through the maximum PI was for Em6+Me2 (8.108) and the lowest value for Em6+Me3 (2.335). Rp ranged from 4.625 (Em5+Me4) to 13.250 (Em6+Me2), with an average value of 8.27. Mean Rp was between 0.272 (Em5+Me4)—0.542 (Em6+Me3) with an average of 0.40. *Liu et al. (2008)* stated higher RP values ranged from 2.229 to 8.457 with an average of 4.927. The fraction of polymorphism, MR, EMR and MI for SRAPs are 1.00, 10.00, 10.00, and 15.65, respectively (Table 6).

## Inter-genotype genetic relationship

Forty-eight pearl millet genotypes were divided into seven major groups by the dendrogram created using pooled data from SSR and SRAP markers based on Jaccard's similarity matrix. Clusters I, II, III, IV, V, VI and VII had one, one, 11, eight , three, 22, and two genotypes, respectively (Fig. 2). In the study of *Nehra et al. (2017)*, with SSR markers, 49 accessions were clustered into eight core clusters. *Kumar et al. (2020b)* alienated 18 lines into three clusters in pearl millet using 74 SSRs. In the current study, the inter-accession genetic coefficient of similarity ranged from 0.616 to 0.877 while the average similarity was 0.743. ICMR 098888 and GHB 905 has a genetic distance (0.384) indicative that both genotypes are having moderate genetic difference level and can be crossed to create a bi-parental mapping population. The minimum genetic distance (0.123) was between IC 139899 and IC 332727, demonstrating that these accessions have more similarity in SSR locus. Moreover, based on diversity results, breeders can select diverse genotypes for combining ability and heterosis analysis for traits studied in the current study.

## CONCLUSION

The genetic diversity for morphological and grain biochemical traits, an outcome of natural selection with the cross-pollination nature of pearl millet, was revealed by analysis of variance. Variability for grain micronutrient content was found greater with a wide range in the population. Genotypes namely ICMR 08666 and IC 139903 were superior for Fe and Zn content, respectively. Genotype ICMR 08666 was also found promising for Zn, K and P content and can further be utilized for genetic biofortification. In the present study, phenotypic diversity analysis grouped all genotypes into nine different clusters. Among all clusters, three clusters had only single genotype with better phenotypic value

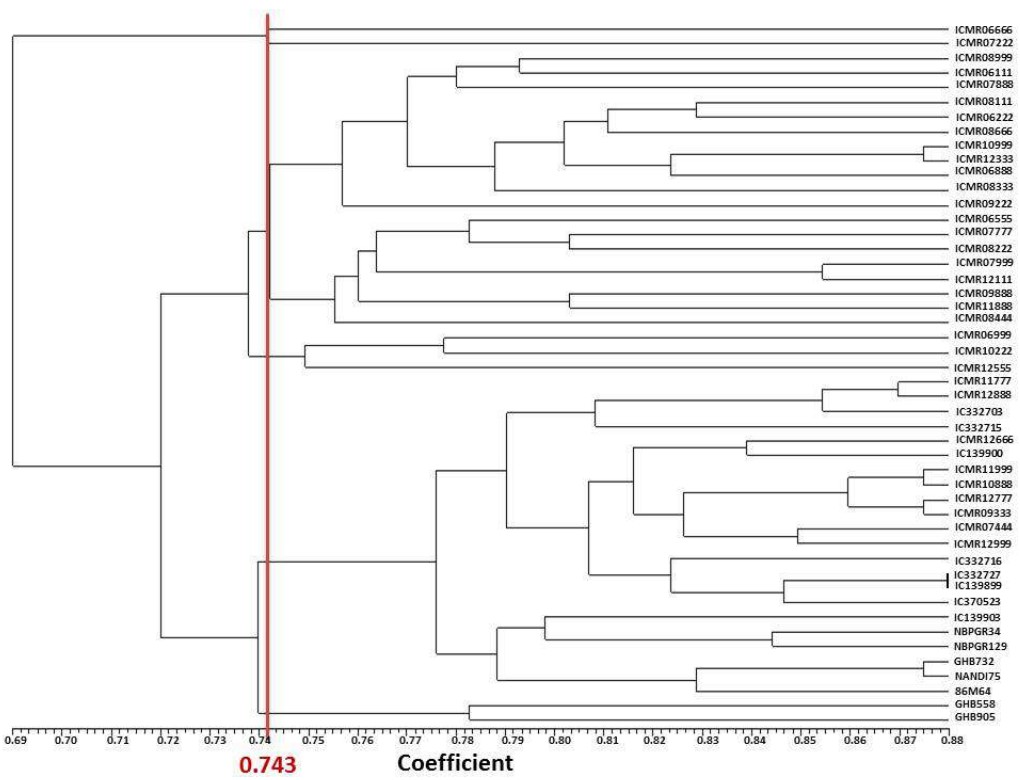

**Figure 2** UPGMA based dendrogram with Jaccard's similarity coefficient of DNA markers.

for most of the grain biochemical parameters. But phenotype is a total outcome of the genotype and its interaction with the environment. Genetic markers are found effective in this study, they help to identify ICMR 098888 and GHB 905 as diverse genotypes for making a bi-parental mapping population.

## ACKNOWLEDGEMENTS

All authors would like to thankful to the Anand Agricultural University for providing the necessary facilities and resources.

### Funding

This work is a Masters thesis work which was not supported by any funding agency. The funders had no role in study design, data collection and analysis, decision to publish, or preparation of the manuscript.

### Competing Interests

Sushil Kumar is an Academic Editor for PeerJ.

## Author Contributions

- Darshanaben F. Gunguniya performed the experiments, prepared figures and/or tables, and approved the final draft.
- Sushil Kumar conceived and designed the experiments, analyzed the data, prepared figures and/or tables, authored or reviewed drafts of the article, and approved the final draft.
- Mukesh P. Patel conceived and designed the experiments, performed the experiments, authored or reviewed drafts of the article, and approved the final draft.
- Amar A. Sakure conceived and designed the experiments, authored or reviewed drafts of the article, and approved the final draft.
- Rumit Patel performed the experiments, analyzed the data, prepared figures and/or tables, and approved the final draft.
- Dileep Kumar performed the experiments, prepared figures and/or tables, and approved the final draft.
- Vikas Khandelwal conceived and designed the experiments, authored or reviewed drafts of the article, provided few genotypes for study, and approved the final draft.

## Data Availability

The raw data are available as Supplemental Files.

## Supplemental Information

Supplemental information for this article can be found online at http://dx.doi.org/10.7717/peerj.15403#supplemental-information.

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
