# Peer review of "Morpho-biochemical characterization and molecular marker based genetic diversity of pearl millet (Pennisetum glaucum (L.) R. Br.)"

_PeerJ, doi:10.7717/peerj.15403_

## Round 0.1 · original submission · Major Revisions

Dear Authors,

The reviewers evaluation process of your manuscript has been completed and the comments have been sent to you. Please take care to pay attention to the comments of the reviewers and the points I have mentioned below while developing your manuscript.

1. Highlight any revisions you have made and thoroughly explain the reasons for disagreeing with the peer review. Pay attention to the rel-upoading rules specified by our journal.

2. I recommend you get proficient speaker support to improve the language of your manuscript. Because the language of your manuscript is very weak and there are many grammatical errors.

3. Pay attention to the more detailed description of your experiment design and methods.

4. Use clearer language and discuss strongly in presenting your findings.

Please, can you send a new and improved version of the manuscript addressing the reviewers' comments?

·

Basic reporting

*Fluent and regular English was not used.
*Literature references provided sufficient field background/context.
*Raw data shared, but insufficient figures

Experimental design

Methods are described with sufficient detail & information, but there is a low number of replicate.

Validity of the findings

*The number of references is sufficient, but there is no fluency in the sentences.

*The conclusion part can be written more impressively.

Additional comments

*A graph in the form of a dendrogram can be drawn on both biochemical data and molecular data, and it should be explained more fluently which genotypes stand out for what purposes.
*English can be reviewed by a fluent speaker.
*In general, the article should be revised with shorter and more concise sentences and a more fluent expression.
Necessary corrections and suggestions were also made on the text.

·

Basic reporting

Dear Editor, the study entitled "Morpho-biochemical characterization and molecular marker-based diversity study in pearl millet [Pennisetum glaucum (L.) R. Br.]" evaluates a total of 48 pearl millet genotypes for morphological, biochemical, and molecular diversity. The study aims to identify superior genotypes for pre-breeding and population generation studies. Due to climate change and need for the improvement in yield and components, it is timely and may be beneficial for the related target groups.

However, there are several major issues in the overall presentation of the study. First is the writing, structure and presentation. It is not presented in smooth and direct language. The almost entire text needs to be edited. I have made a lot of grammatical and other language-related corrections but it was not enough. Please find my specific comments, queries, and edits in the attached file.

Experimental design

Is the experiment repeated only once with 2 replications? Would this be enough to reduce the environmental effects? So, how would these results be applied to other seasons/environments/conditions?

Please find my specific comments, queries, and edits in the attached file.

Validity of the findings

The results seem valid to me. However, they are not well discussed with the relevant studies. Discussion needs to be improved for language and for the presentation

Reviewer 3 ·

Basic reporting

Generally is a good work. Congratulation. There are some gramatical errors, ı indicated them.

line 35. R. Br.]c ?
Line 38 and 39. india and Africa should be replaced instate of indian and african
Line 57. minor changes
Line 59. Fe and Zn names
Line 91-92. Normallly replication should be at least 3. But because of the good morphological data and molecular analysis, it is acceptble.
Line 95. Is the table 1 and supp table 1 same? If it is yes please indicate it and combine or if it not please put the supp table 1
I suggest not only

Experimental design

Line 91-92. Normallly replication should be at least 3. But because of the good morphological data and molecular analysis, it is acceptble.

Please explain the importance of the 8 parameters and how you obtain these parameters datas
(eight morphological traits viz., days to 50% flowering, plant height, panicle diameter (PD, mm), panicle length (PL, cm), number of productive tillers, grain yield, days to maturity, 1000- grain weight ).

Please cite or give a reference or a little information about "Vernier calipers".
Line 136-137. variance (ANOVA) (Which statistical software did you use?)
Line 144-145. UPGMA (Long name? dendrogram.
Line 149.No need to put the ANOVA title. Just give the statistical results with their importance grade, significant levels,
Line 344-345-346-347. Figure 1 not avaliable. Please put in it. At the end of the manuscript.
Line 383. This reference not cited in main text.

Validity of the findings

I believe that it contributes more to the development of science by thinking about the issues that found different results and why the differences between our results and ours might have arisen, rather than discussing with articles that found similar or very close results to the findings in this article. Although finding similar results does not contribute to science, the article is well-structured. It has been a nice study with phenotypic and genotypic data.

Additional comments

I did the comments, changes changes marked in red colour.
I highly recommed you that the nutritional content of this plant, its properties, etc. Numerous publications have been made about its properties. I think that it will be more beneficial for people to focus on concrete results for breeding, developing, processing and selling this plant, which is highly preferred by people in semi-arid regions, and selling it to the countries of the world and making it a source of livelihood for people.

Annotated reviews are not available for download in order to protect the identity of reviewers who chose to remain anonymous.

---

## Round 0.2 · Minor Revisions

Dear authors,

It seems that there are still missing points in the revisions requested by the reviewers. Please review them carefully, change them and resubmit. Do not forget to include a letter including answer the reviewers regarding the changes you have made.

·

Basic reporting

None

Experimental design

None

Validity of the findings

None

Additional comments

The corrections given earlier were not made to the extent expected. They need to be reviewed in more detail. His English is still weak and not fluent. There is a lot of detail in my writing. The sentences need to be shortened.

·

Basic reporting

I have re-evaluated the revised version of the manuscript. Even though it has some improvements, it still has several structural and grammatical issues. The Ms. also needs to be edited for language. Please find my specific comments in the attached Ms. file with comments and some track-changed corrections.

Experimental design

Please find my specific comments in the attached Ms. file with comments and some track-changed corrections.

Validity of the findings

Please find my specific comments in the attached Ms. file with comments and some track-changed corrections.

Additional comments

Please find my specific comments in the attached Ms. file with comments and some track-changed corrections.

Reviewer 3 ·

Basic reporting

All the revisions ı have suggested has been done. From my side it is ok.

Experimental design

Experiment design is good. As ı indicated in revision file, normally a pattern of experiment with 2 replication not accepted, but becau of the genotypes are large, enriching the study with morphological, biomolecular analysis, as a results, this manuscript can be accept to be published.

Validity of the findings

No comment.

Additional comments

Normally a pattern of experiment with 2 replication not accepted, but becau of the genotypes are large, enriching the study with morphological, biomolecular analysis, as a results, this manuscript can be accept to be published.

---

## Round 0.3 · Minor Revisions

Dear Authors,

I see that there are still shortcomings in the last changes you made. As an editor, I made some corrections.

1) I would like you to review the corrections in the text on the attached file I sent you.

2) Also review the references section and correct any deficiencies.

3) Please use abbreviations throughout the text when referring to element contents. It was enough to give the long spelling once, you've already explained it in parentheses. If you have given abbreviations for the other traits you have studied, please give the abbreviation to ensure uniformity throughout the text.

4) Change the title of Table 1 as "Analysis of variance (ANOVA) of studied traits in pearl millet".

5) Change the title of Table 2 as "Descriptive statistics for studied traits in pearl millet" and write the explanations of the abbreviations "SEm, CD@5% and CV" under the table.

6) More details on Table 3 please. Why is there a grouping by DFF and TGW? How accurate is it to explain agronomic performance with just these two features? Explain or delete Table 3 and its reference in the text.

7) If you delete Table 3, also rearrange the numbers of subsequent Tables and their references in the text.

8) Include the dendogram of your Cluster analysis for the phenotypic data. Since this would be Figure 1 in this case, correct Figure 1 as Figure 2 and edit the citations in the text as well.

---

## Round 0.4 · accepted · Accept

The changes we suggest have been made properly and are appropriate for publication. Congratulations. Good luck in your new researches.